# Perception Matters: Perceived vs. Objective Air Quality Measures and Asthma Diagnosis among Urban Adults

**DOI:** 10.3390/ijerph20176648

**Published:** 2023-08-25

**Authors:** Jane E. Clougherty, Pilar Ocampo

**Affiliations:** 1Department of Environmental & Occupational Health, Drexel University Dornsife School of Public Health, Philadelphia, PA 19104, USA; 2Johns Hopkins Bloomberg School of Public Health, Baltimore, MD 21205, USA; pocampo2@jhu.edu

**Keywords:** perceived air quality, environmental stress, psychosocial pathways, urban air quality, adult asthma

## Abstract

Urban air pollution is consistently linked to poorer respiratory health, particularly in communities of lower socioeconomic position (SEP), disproportionately located near highways and industrial areas and often with elevated exposures to chronic psychosocial stressors. Fewer studies, however, have considered air pollution itself as a psychosocial stressor and whether pollution may be impacting health through both direct physiologic and psychosocial pathways. We examined data on perceived air pollution exposures from a spatially representative survey of New York City adults through summer and winter 2012 (*n* = 1183) using residence-specific ambient nitrogen dioxide (NO_2_) and fine particulate matter (PM_2.5_) exposure estimates. We used logistic regression to compare associations for perceived and objective air quality on self-reported asthma and general health, adjusting for sociodemographics and mental health. In models including all exposure metrics, we found small but significant associations for perceived air quality (OR = 1.12, 95% CI: 1.04–1.22) but not for NO_2_ or PM_2.5_. Neither perceived nor objective pollution was significantly associated with self-reported general health. Results suggest that perceived air quality may be significantly associated with adult asthma, more so than objective air pollution and after adjusting for mental health—associations not observed for self-reported general health.

## 1. Introduction

Substantial evidence in environmental epidemiology links urban air pollution to chronic disease outcomes, including respiratory and cardiovascular disease, mortality, and cognitive and behavioral outcomes [1]. Traditionally, however, the environmental psychology and human geography literature has conceptualized environmental conditions as a human experience with both physical and psychosocial dimensions [2]. Models and frameworks have been developed to examine the “environmental stress” of living in an environment that one believes to be harmful [3], and the field of environmental psychology has demonstrated profound interactions between the environment, human perception, and human well-being, particularly mental and emotional well-being. Historically, poor air quality has been associated with outcomes as varied as behaviors [4] and aggression [5], test performance [6], neurologic impacts [6], vision impairment [7], and psychological effects [8,9] including vigilance [10,11] and anxiety [12]. These outcomes have been attributed, to varying degrees, to the direct physical impacts of pollution or to psychosocial pathways stemming from perceived pollution exposures [5].

Health impacts of perceived pollution exposures have been examined under a framework referred to as environmental stress, wherein individuals may experience distress related to fear or worry about a perceived environmental contamination, which may then be manifested in a range of psychological, social, or behavioral outcomes [3,13]. Health geographers influenced by environmental justice research have long argued that pollution sources differentially located in lower-income communities (e.g., highways, industry, and smokestacks) are visual cues that can signal to residents that their health and well-being is devalued by the larger society [14]. Accordingly, health geographers have explored the portion of a given source’s health impact that may be attributable to a sense of anger, frustration, or disenfranchisement in communities near industrial facilities in southern Ontario, Canada, and elsewhere [15,16,17]. The work is in keeping with classic research on the “broken windows” theory, which posits that physical disorder and disinvestment indicate to residents that their well-being is less valued than that of residents in other communities, producing consequent health effects via stress-related and behavioral pathways [18]. The work also demonstrates the challenge of disentangling direct physical impacts of pollution on human health from those attributable to worry or frustration about perceived exposures, particularly in communities near toxic sites [15,16,17].

A few studies have compared the health effects attributable to the physical vs. psychosocial aspects of a given source. For example, in a study of individuals living near a hazardous waste site in southern California, worry about health effects was shown to greatly increase symptom prevalence over that which would be predicted by the pollution exposure alone [19]. Other studies have shown psychosocial effects to be stronger than objective exposure, where, for example, worry about health effects of an industrial source better predicted illness than did objective measures of exposure [20,21]. Similarly, in an experimental model, Dalton and colleagues informed participants that an odorous exposure was either hazardous or beneficial, finding that the hazardous belief produced significantly more symptoms [22,23]. Finally, Claeson et al. performed a path analysis of perceived and objective pollution measures, health risk perception, annoyance, and symptoms among residents of an industrial region in Sweden, finding that symptoms were not directly attributable to objective pollution but were mediated via perceived pollution health risk [24].

This latter finding has been reaffirmed by other studies in Europe, including highly-industrialized areas of Estonia and Sweden [25,26], and supports a model in which the belief that an exposure is harmful (or that one is susceptible to said exposure) itself leads to subsequent psychologic or physiologic sequelae. Accordingly, odor, snow discoloration [27], or other indicators of poor air quality may provide sensory cues that facilitate pollution- and/or stress-related illness, symptoms, or attention to symptoms [19]. This model reaffirms the health geography perspective that perceived air pollution exposure is only one part of the more holistic experience of place, as neighborhood residents may experience the context or quality of a community as a whole [2] rather than drawing distinctions among specific social or physical environmental conditions (as is our norm in epidemiology). For a few distinct environmental exposures, such as odor from hog operations or occupational noise, researchers have considered discomfort or annoyance to be almost inherent aspects of the exposure experience, and they possibly serve as those strong mediators of health effects [19,24,28].

Only in recent decades has environmental epidemiology begun to seriously explore interactions between physical pollutant exposures and psychosocial experience [29,30]. This interest stems from the recognition that pollution sources are often co-located with social and psychosocial stressors (e.g., poverty and violence), to the extent that lower-income and minoritized communities also experience poorer air quality. Growing evidence from social epidemiology and psychoneuroimmunology demonstrate that chronic stress can profoundly shape health and susceptibility through immune, endocrine, metabolic, and other pathways—collectively known as “allostatic load” [31]. This work has brought psychosocial pathways into serious consideration in environmental epidemiology and challenged epidemiologists to think differently about how we quantify those effects hypothesized to be operating via human perception and appraisal vs. those that we hypothesize to operate via traditional physiologic exposure pathways (i.e., inhalation, ingestion, and dermal contact).

Given this growing interest in the combined impacts of social and physical exposures on health, it is pertinent to ask whether some pollution sources may be operating via both physical and psychosocial pathways simultaneously and what portion of any resultant health impacts may be operating through each. Evidence indicates that residents can detect air pollution even below regulatory guidelines with resultant negative health effects [32] and that residents perceive environmental exposures to be part of the context of a place [2], which is closely intertwined with neighborhood identification, stigma, and perceptions of abandonment, lack of control, or fear of displacement [33]. 

Though perceived air quality has been strongly correlated with objective air quality in some ambient settings—for NO_2_ [32,34,35,36,37], PM_2.5_ [37,38], and SO_2_ [35] and PM sulfur content [39]—these relationships are not entirely consistent [1,40], and are shown to vary by individual-level characteristics including disease status (especially for respiratory illness or perceived susceptibility) [35,36,37,39,41,42,43,44], gender (with females generally perceiving worse air quality) [32,35,37,38,39,43,45], age (with younger adults perceiving worse air quality) [1], socioeconomic position (with, conversely, more highly educated persons [1,39,40,44] or those living in areas of greater material deprivation [32,40,46] perceiving worse air quality), smoking status (with smokers perceiving less pollution or pollution-related annoyance, [47] and non-smokers exposed to environmental tobacco smoke perceiving higher concentrations [39]), pollution site characteristics [15], personal affect [48], and awareness of environmental exposures and risks [16,17]. Interestingly, early work on perceived vs. objective pollution exposures in the 1970s indicated lesser perceived exposure and risk among Black (compared to White) Americans [49], although this relationship has shifted in recent decades with greater awareness of issues of environmental injustice [45,46,50]. This variance may call into question the use of perceived pollution as a simple proxy for objective exposures despite the demonstrated utility of validated scales to measure perceived air quality [32,51].

In this paper, we examine and compare associations for “perceived” and “objective” measures of urban air pollution with self-reported asthma status and self-related health among adult residents of New York City (NYC). Our primary hypotheses are as follows: (1) that perceived air quality more strongly predicts asthma status than does objective air quality, after accounting for potential confounders; (2) this association is robust to adjustment for mental health and affect, which may affect air quality appraisal; and (3) this association does not hold for overall self-reported health, which may be less directly influenced by air pollution. To test these hypotheses, we used the highly refined land-use use-regression models of the New York City Community Air Survey (NYCCAS) [52,53] and spatially distributed citywide survey data on perceived neighborhood air quality, chronic stress, and self-reported physical and mental health conditions. We used conditional logistic regression, adjusting for sociodemographic confounders and examining associations for perceived and objective pollution exposure metrics, both separately and combined. To account for same-source bias (i.e., individuals reporting on both their own perceived exposures and health) and reverse causation (i.e., individuals with poor health (asthma) may perceive worse local air quality), we adjusted all models for self-reported general health, the presence of any mental health condition, and depression. Finally, as a negative control, we examined comparable models predicting self-reported general health as the outcome, which we hypothesize to be less directly impacted by air quality and which residents may be less inclined to associate with air quality, thereby presenting less same-source bias in keeping with the observation reported above that individuals with respiratory illness or other specific perceived susceptibility to air pollution are shown to perceive air quality more negatively [35,36,37,39,41,42,43,44].

## 2. Materials and Methods

### 2.1. Data Source

This study combined spatially refined air quality data with data on perceived air quality, perceived stress, and mental health from a citywide spatially representative survey of NYC adults (*n* = 1528). We implemented a triple-framed survey with inclusion criteria of current NYC residency, over 18 years of age, speaking either English or Spanish, and using random digit dial (RDD) of NYC landline (*n* = 319) and cellular (*n* = 85) phone numbers. In households with multiple eligible adults, we surveyed only the adult with the most recent birthday. An additional 779 adults in a voluntary, standing survey panel of NYC adults completed a self-administered survey through an online platform (Survey Sampling International, http://www.surveysampling.com, accessed on 24 August 2023).

Within each sampling frame, we aimed for a spatially representative sample, setting a priori sampling targets by percent of NYC residents in each borough (~17, 31, 20, 27, and 5% from the Bronx, Brooklyn, Manhattan, Queens, and Staten Island, respectively). The survey was conducted in summer (June–September 2012) and winter (December–March 2012–2013) seasons with independent RDD samples to balance seasonal differences in mood and psychological well-being. Participants were offered a USD 10 incentive to complete the approximately 30 min survey.

RDD telephone surveys were administered by trained interviewers at the Survey Research Program of the University Center for Social and Urban Research (UCSUR) at the University of Pittsburgh using computer-assisted telephone interview (CATI) software. Interviewers assessed participant eligibility, provided the option to complete the survey in English or Spanish, and obtained informed consent prior to survey initiation. The order of survey scales was fixed to avoid priming and to optimize flow by grouping scales with similar response options. The order of items within scales was randomized where appropriate. Surveys were implemented under the oversight of the University of Pittsburgh Institutional Review Board protocol #PRO10110530.

### 2.2. Perceived Air Quality Metric

The citywide stress survey included three items on perceived neighborhood air quality, adapted from Forsberg et al., as developed for a study of perceived environmental quality in Sweden [32]: (Q1), “The air in my neighborhood seems worse than in other neighborhoods”; (Q2), “I am bothered by pollution from cars, trucks, or buses in my neighborhood”; and (Q3), “I am bothered by air pollution from industry or other pollution sources in my neighborhood”. Each response was scored on a 4-point Likert scale: 1—strongly disagree, 2—disagree, 3—agree, and 4—strongly agree. A total perceived air quality score was created by summing the three responses, with scores of 4–6 indicating relatively good perceived air quality and scores above 6 indicating relatively poor perceived air quality. For logistic regression analysis, we created a binary variable, with summed scores above the median (greater than 6) indicating poor perceived air quality.

### 2.3. Objective Air Quality Estimates

Objective air quality was estimated as average concentrations of NO_2_ and PM_2.5_ within 300 m of the residential address, using 100 m resolution surfaces provided by the New York City Community Air Survey (NYCCAS) and performed by NYC Department of Health and Mental Hygiene (DOHMH), which monitored 155 sites across NYC for multiple pollutants year-round from December 2008–November 2010 [52,53]. All NYCCAS study methods, including sampling design, quality control and quality assurance methods, and modeling approaches, are detailed elsewhere [52,53]. Briefly, sites were selected by stratified random sample to capture source variation (i.e., traffic, buildings) while ensuring spatial coverage. Two-week samples were collected from spatially distributed sites once per season. Five “reference” sites were sampled every session. Pollutants were selected to capture key local sources in NYC potentially amenable to policy intervention. Here, we focus only on PM_2.5_ and NO_2_, which are criteria pollutants with somewhat different spatial variation in NYC and primarily associated with oil burning and vehicular traffic, respectively. Land-use regression (LUR) and kriging with external drift were used to derive 100 m grid surfaces [52], and we averaged concentrations for all grid centroids within 300 m of each self-reported residential location. (Only the nearest cross-street was requested in the survey for anonymity).

### 2.4. Physical and Mental Health Outcomes Data

Prevalence of asthma among adult survey respondents was assessed using the question, “Have you ever been told by a healthcare provider that you have asthma?”. Self-reported general health was assessed using the question, “Would you say that in general your health is excellent, very good, good, fair or poor?”.

We identified participants with mental health challenges as having any of the following:(1)An answer of “yes” to either of two questions (“During the past year, have you received any counseling or taken prescription medication for your mental health?” or “Have you ever been told by a doctor, nurse or other health professional that you have depression?”);(2)A score indicating “high” stress on the 4-item Perceived Stress Scale (PSS) [54];(3)A score indicating clinical concern on either the 23-item MMPI-2 Anxiety Scale (ANX) [55] and the 10-item CES-D Depression Scale (DEP) [56,57].

To reduce participant burden, we used the shortest validated version of each instrument. Because the PSS contains only four items, we imputed conservatively; for participants with only one skipped or refused item on a given scale, we imputed using the mean for all other responses after reverse-coding. In all cases, non-respondents differed little (<5%) from the overall sample population on average.

### 2.5. Data Analysis

We used Pearson correlations to assess bivariate associations among exposure metrics. Logistic regression was used to determine strength of association between exposures and two self-reported health outcomes: asthma (binary: yes/no ever diagnosed) and self-reported general health (categorical: five categories, excellent to poor).

Ten models were developed in total: five per outcome. Models 1–3 included one exposure variable (NO_2_, PM_2.5_, or Perceived Pollution Score) but were adjusted for all individual-level confounders detailed in Table 1 (sex, race/ethnicity, age, education, insurance status, income, marital status, family size, neighborhood tenure, season, and survey frame).

Model 4 included all three exposure metrics (mutually adjusted), and model 5 was further adjusted for measures of mental health diagnosis and depression, which may plausibly confound the relationship between perceived environmental quality and self-reported physical health outcomes. Due to missing data, only 1183 participants with complete data were included in all final models for comparability across model results; participants excluded due to missing data did not significantly differ on key exposures or outcomes.

Models were sensitivity-tested using alternative model forms (Proc GLM), varying the resolution of confounder metrics (e.g., categorical vs. continuous where possible), varying the threshold for mental health outcomes and composition of the mental health challenge variable, and using categorical forms of outcome metrics where possible.

All analyses were performed in SAS v 9.4 (Cary, NC, USA). Logistic regression was performed using Proc Logistic Descending.

## 3. Results

### 3.1. Sample Characteristics

In the sample population, 60% of participants were longer-term residents, having lived in their neighborhood for 10 years or more. Forty-five percent had incomes below USD 46,000 per year, and 46% held at least a bachelor’s degree, but only 10% reported having no health insurance (Table 1).

### 3.2. Objective and Perceived Air Pollution Exposures

Estimated exposures varied substantially across the cohort—ranging from 12.6–47.6 ppb for NO_2_ and 8.6–16.3 µg/m^3^ for PM_2.5_.

Among survey respondents, 19% believed the air in their neighborhood seemed worse than in others (i.e., “agreed” or “strongly agreed” with Question 1). Further, 30% were bothered by air pollution from cars, trucks, or buses (Q2), and 20% were bothered by air pollution from industry or other pollution sources (Q3). Most (68%) of the survey respondents had a total perceived pollution score of 6 or lower, indicating that most respondents were not greatly bothered by neighborhood air pollution.

Although the perceived pollution items were strongly correlated among themselves (r = 0.66–0.90) (Table 2), they were weakly correlated with objective air quality metrics (r = 0.06–0.12).

We found a significant association between perceived air pollution and self-reported asthma diagnosis even after adjusting for self-reported general health and all potential confounders (Table 3: model 1.3: adj OR = 1.14 (95% CI = 1.05–1.23)). Notably, this association for perceived air pollution also held true after adjusting for objective NO_2_ and PM_2.5_ exposures in the same model (model 1.4: adj OR = 1.13 (1.04–1.23) vs. model 1.3: adj OR = 1.14 (1.05, 1.23)). In contrast, we found no significant associations between objective air quality and self-reported asthma (models 1.1, 1.2, 1.4, and 1.5), and both associations moved further towards the null after including perceived pollution in the same model. These results support the interpretation that perceived pollution exposures may be a stronger and more robust predictor of asthma than are objective PM_2.5_ or NO_2_ concentrations.

In contrast, self-reported general health was not significantly associated with any of the air pollution exposure metrics, either objective or perceived (Table 4). Better self-reported general health was inversely associated with co-morbidity of asthma, depression, or other mental health diagnosis. Better general health was also associated with younger age, higher incomes, Asian ethnicity, and during winter season. Having Medicare or no health insurance conferred better, higher self-reported general health.

## 4. Discussion

We found a significant, positive association between perceived pollution and self-reported asthma status, which was robust to extensive adjustment for individual-level confounders and to adjustment for both objective fine-scale air pollution exposures and mental health/affect. These results suggest that perceived air pollution exposures may be a stronger and more robust predictor of asthma than are objective NO_2_ and PM_2.5_ concentrations, in some cases.

In contrast, using general health as a negative control, we found no significant association between perceived air quality and overall self-reported health.

These results suggest a meaningful role for the perception of outdoor air quality in facilitating its impacts on respiratory health. In contrast, for those broader general health conditions not normally associated with air quality by community residents, as captured by self-reported general health status, we saw no such association.

These results are important for the following reasons: (1) robustness to adjustment for objective air quality suggests that perceived pollution is more than a simple proxy for physical pollution exposure but rather is capturing additional dimensions relevant to respiratory health; (2) robustness to adjustment for mental health/affect suggests that associations between perceived pollution and health are not solely a product of reverse causation (persons of lower affect more negatively appraising environmental conditions); and (3) the lack of association with self-reported general health suggests some specificity to outcomes broadly accepted to be influenced by air quality (i.e., respiratory health).

These observations are supported by a study in health geography, which has long documented adverse health effects of perceived pollution exposures, which are traditionally referred to as “environmental stress” [3,13]. Prior studies have shown that objective and perceived air pollution are generally positively correlated [32,34,35,36,37,38,39], particularly for odorous or visible pollutants [19,27]. There are, of course, many nuances in this relationship, and perceived environmental quality is shown to vary by race/ethnicity and socioeconomic position (SEP). Our findings are also consistent with studies that have found asthma to be significantly predicted by negative perceived air quality [36,58]. Similarly, Elliot et al. [59] found the relationship between physical air pollution and health to be significantly mediated by perception of air pollution exposures.

Our findings are also in keeping with a growing body of literature on perceived indoor air quality and health. Poor perceived indoor air quality has been associated with worse worker cognitive performance, physiological parameters [2], and health [3]. Similar results have also been identified for perceived home indoor air quality and health in both children [4] and adults [5].

Consistent with other studies, we did find a positive association between asthma and poor mental health, although this association did not fully mediate relationships between perceived air pollution and asthma. Piro et al. [36] reported that people with non-respiratory-related chronic disease, including mental health conditions, tended to over-report air pollution problems. We also found significant associations between objective NO_2_ and PM_2.5_ exposure estimates and mental health conditions and depression, consistent with the growing body of literature linking air pollution to mental and cognitive impacts. For example, a study by Cho et al. [60] found air pollution to be associated with emergency department admissions for depression. Niu et al. found higher risk of mental-health-related ED and hospital encounters among children on high-temperature days [61].

### 4.1. Strengths

Our study benefits from a number of strengths, including a systematic citywide survey designed to capture perceived air quality, stress, and mental health across all NYC neighborhoods and a previously validated instrument to capture perceived pollution exposures. In addition, we used very fine-scale air pollution exposure estimates (100 m resolution) to estimate residence-specific chronic exposures to multiple air pollutants. In addition, we were able to adjust, within these data, for many individual-level risk factors that may otherwise be posited to mediate perception–health relationships (e.g., including income, race/ethnicity insurance status, etc.).

Our approach leveraged aspects of both mediation analysis (accounting for mental health as plausible mediator of perceived air-quality–health relationships) and a negative control (i.e., the lack of association with self-reported general health, less commonly believed to be associated with ambient air quality in contrast to asthma).

### 4.2. Limitations

Despite these notable strengths, our study also has several important limitations. Our survey was, unfortunately, under-powered for subgroup analysis; it would be interesting in the future to examine whether these associations vary by gender, race/ethnicity, age, or comorbidities. Self-report bias likewise remains a challenge in assessing the relationship between a perceived exposure and self-reported asthma, although our results are bolstered by our ability to adjust for potential mediation by mental health and the negative control analysis revealing null associations with self-reported general health.

## 5. Conclusions

We found that adult asthma was more strongly associated with perceived than objective air pollution exposure estimates even after adjustment for sociodemographic variables and mental health indicators—associations that did not hold for self-reported general health.

Our results suggest that urban air pollution may impact health through psychosocial as well as physical pathways. This annoyance, i.e., the negative perception of local air quality, may be a chronic stressor that directly contributes to disease risk. As such, attention to environmental concerns of disproportionately numerous sites of hazardous waste facilities in or near lower-income and minoritized communities may take on additional meaning, as the presence of polluting sources may signal to residents a devaluing of the local neighborhood and resident health, directly impacting health over and above the impact of physical pollution exposures.

## Figures and Tables

**Table 1 ijerph-20-06648-t001:** Participant Characteristics and Potential Factors included in Logistic Regression Analysis.

	Count (*n* = 1183)	Percent (%) of Total		Count (*n* = 1183)	Percent (%) of Total
Sex			Age Mean (Std Dev)		
Female	745	62.98	44.3 (±17.0)	1183	100
Male	432	36.52	Primary Insurance		
Do not know/refused	6	0.51	Private	581	49.11
Race			Medicare	179	15.13
White	634	53.59	Medicaid	176	14.88
Black	265	22.4	Self-pay	29	2.45
Asian/Pacific	122	10.31	Other	56	4.73
Native American	21	1.78	No insurance	124	10.48
Other	123	10.4	Do not know/refused	38	3.21
Do not know/refused	18	1.52	Income		
Hispanic/Latino			Less than USD 23,000	269	22.74
Hispanic/Latino	231	19.53	USD 23,000–USD 46,000	260	21.98
Non-Hispanic/Latino	941	79.54	USD 46,000–USD 70,000	212	17.92
Do not know/refused	11	0.93	USD 70,000–USD 93,000	168	14.20
Years Lived in Neighborhood			USD 93,000–USD 135,000	123	10.40
Less than 1 year	80	6.76	USD 135,000 or more	117	9.89
1–5 years	231	19.53	Do not know/refused	34	2.87
5–10 years	173	14.62	Marital Status		
>10 years	699	59.09	Married/living as married	445	37.62
Education			Separated	22	1.86
Less than 8th	10	0.85	Divorced	106	8.96
Some high school	47	3.97	Widowed	73	6.17
High school/GED	175	14.79	Single/never married	531	44.89
Trade/vocational/nursing	25	2.11	Do not know/refused	6	0.51
Some college, no degree	273	23.08	Survey Season		
Associate’s	93	7.86	Summer	460	38.88
Bachelor’s	325	27.47	Winter	723	61.12
Master’s	169	14.29	Survey Format		
Professional’s	37	3.13	Cell	85	7.19
Doctoral	26	2.2	Landline	319	26.97
Do not know/refused	3	0.25	Online	779	65.85

**Table 2 ijerph-20-06648-t002:** Pearson correlations (*p*-value) among objective and perceived air pollution metrics (*n* = 1183).

	NO_2_	PM_2.5_	Perceived Pollution Q1	Perceived Pollution Q2	Perceived Pollution Q3
NO_2_					
PM_2.5_	0.91				
	(<0.0001)				
Perceived Pollution Q1	0.09	0.08			
	(0.0022)	(0.0052)			
Perceived Pollution Q2	0.12	0.11	0.66		
	(<0.0001)	(0.0002)	(<0.0001)		
Perceived Pollution Q3	0.06	0.06	0.69	0.69	
	(0.0488)	(0.043)	(<0.0001)	(<0.0001)	
Perceived Pollution Sum	0.10	0.09	0.88	0.89	0.90
	(0.0006)	(0.0014)	(<0.0001)	(<0.0001)	(<0.0001)

**Table 3 ijerph-20-06648-t003:** Logistic Regression Models for Self-Reported Asthma Diagnosis.

	Model 1.1: NO_2_		Model 1.2: PM_2.5_		Model 1.3: Perceived Pollution		Model 1.4: All Exposures	
	OR	(95% CI)	OR	(95% CI)	OR	(95% CI)	OR	(95% CI)
Potential Exposures of Interest								
NO_2_	1.16	(0.81, 1.65)	-	-	-	-	1.03	(0.44, 2.38)
PM_2.5_	-	-	1.64	(0.46, 5.90)	-	-	1.39	(0.07, 29.08)
Perceived pollution (PP)	-	-	-	-	**1.14**	**(1.05, 1.23)**	**1.13**	**(1.04, 1.23)**
Self-Reported Health								
Excellent	1		1		1		1	
Very good	1.00	(0.53, 1.90)	1.00	(0.53, 1.90)	1.07	(0.56, 2.04)	1.06	(0.55, 2.02)
Good	**2.47**	**(1.35, 4.50)**	2.47	(1.36, 4.52)	**2.63**	**(1.43, 4.84)**	**2.64**	**(1.43, 4.85)**
Fair	**3.23**	**(1.64, 6.38)**	**3.24**	**(1.64, 6.40)**	**3.45**	**(1.74, 6.87)**	**3.44**	**(1.73, 6.84)**
Poor	2.33	(0.76, 7.09)	2.31	(0.76, 7.05)	2.48	(0.81, 7.56)	2.47	(0.81, 7.55)
Mental health condition “Yes”	1.23	(0.74, 2.06)	1.24	(0.74, 2.07)	1.31	(0.78, 2.19)	1.31	(0.78, 2.21)
Depression “Yes”	**2.27**	**(1.43, 3.60)**	**2.27**	**(1.43, 3.60)**	**2.13**	**(1.34, 3.39)**	**2.10**	**(1.32, 3.35)**
Sex								
Male (Reference)	1		1		1		1	
Female	**0.62**	**(0.43, 0.89)**	**0.62**	**(0.43, 0.89)**	**0.64**	**(0.44, 0.92)**	**0.64**	**(0.44, 0.92)**
Do not know/refused	0.89	(0.09, 8.98)	0.89	(0.09, 8.96)	1.05	(0.10, 10.67)	1.11	(0.11, 11.37)
Race								
White (Reference)	1		1		1		1	
Black	**1.90**	**(1.21, 2.97)**	**1.89**	**(1.21, 2.96)**	**1.96**	**(1.25, 3.08)**	**1.97**	**(1.25, 3.10)**
Asian	0.60	(0.28, 1.26)	0.61	(0.29, 1.28)	0.59	(0.28, 1.24)	0.59	(0.28, 1.25)
Native	2.11	(0.64, 6.99)	2.10	(0.63, 6.95)	1.96	(0.59, 6.49)	1.95	(0.59, 6.45)
Other	0.90	(0.46, 1.74)	0.89	(0.46, 1.73)	0.96	(0.49, 1.88)	0.95	(0.49, 1.87)
Do not know/refused	1.13	(0.26, 4.97)	1.11	(0.25, 4.89)	1.13	(0.26, 5.01)	1.13	(0.25, 5.05)
Hispanic/Latino Ethnicity								
No (Reference)	1		1		1		1	
Yes	**2.02**	**(1.23, 3.31)**	**2.01**	**(1.22, 3.29)**	**1.97**	**(1.19, 3.25)**	**1.96**	**(1.19, 3.24)**
Do not know/refused	0.69	(0.08, 5.94)	0.69	(0.08, 5.90)	0.67	(0.08, 5.95)	0.67	(0.08, 5.91)
Age (years)	**0.98**	**(0.96, 0.99)**	**0.98**	**(0.96, 0.99)**	**0.98**	**(0.96, 1.00)**	**0.98**	**(0.96, 1.00)**
Time in neighborhood (decades)	**1.25**	**(1.02, 1.54)**	**1.26**	**(1.02, 1.55)**	**1.25**	**(1.01, 1.54)**	**1.25**	**(1.02, 1.54)**
Household size	1.05	(0.92, 1.19)	1.05	(0.92, 1.19)	1.04	(0.92, 1.18)	1.05	(0.92, 1.19)
Education								
Less than 8th (reference)	1		1		1		1	
Some high school	0.89	(0.12, 6.67)	0.90	(0.12, 6.77)	0.78	(0.11, 5.84)	0.84	(0.11, 6.36)
High school/GED	2.14	(0.36, 12.86)	2.16	(0.36, 13.08)	1.99	(0.33, 11.82)	2.12	(0.35, 12.86)
Trade/vocational/nursing	1.40	(0.17, 11.38)	1.43	(0.17, 11.76)	1.40	(0.17, 11.25)	1.50	(0.18, 12.39)
Some college	1.39	(0.23, 8.27)	1.41	(0.23, 8.47)	1.39	(0.23, 8.23)	1.47	(0.24, 8.87)
Associate’s	1.50	(0.23, 9.66)	1.53	(0.24, 9.90)	1.39	(0.22, 8.85)	1.48	(0.23, 9.64)
Bachelor’s	2.11	(0.35, 12.93)	2.15	(0.35, 13.24)	2.12	(0.35, 12.95)	2.22	(0.36, 13.76)
Master’s	2.53	(0.40, 15.88)	2.56	(0.41, 16.16)	2.46	(0.39, 15.39)	2.59	(0.41, 16.46)
Professional’s	3.12	(0.41, 23.75)	3.15	(0.41, 24.07)	3.21	(0.42, 24.43)	3.31	(0.43, 25.50)
Doctoral	2.50	(0.29, 21.42)	2.56	(0.30, 22.01)	2.48	(0.29, 21.25)	2.57	(0.30, 22.37)
Do not know/refused	4.61	(0.22, 95.19)	4.66	(0.22, 96.86)	4.15	(0.20, 86.23)	4.41	(0.21, 92.42)
Income								
Less than USD 23,000	1		1		1		1	
USD 23,000–USD 45,999	1.03	(0.60, 1.77)	1.03	(0.60, 1.77)	1.04	(0.61, 1.79)	1.05	(0.61, 1.81)
USD 46,000–USD 69,999	0.85	(0.45, 1.61)	0.85	(0.45, 1.61)	0.91	(0.48, 1.73)	0.93	(0.49, 1.75)
USD 70,000–USD 92,999	1.11	(0.56, 2.20)	1.11	(0.56, 2.19)	1.19	(0.60, 2.36)	1.19	(0.60, 2.37)
USD 93,000–USD 135,000	1.25	(0.60, 2.63)	1.25	(0.59, 2.63)	1.43	(0.67, 3.03)	1.44	(0.68, 3.06)
USD 135,000 +	1.26	(0.54, 2.91)	1.26	(0.54, 2.91)	1.45	(0.62, 3.38)	1.43	(0.61, 3.35)
Do not know/refused	0.30	(0.06, 1.45)	0.30	(0.06, 1.46)	0.35	(0.07, 1.65)	0.33	(0.07, 1.59)
Marital Status								
Married/living as married	1		1		1		1	
Separated	0.29	(0.04, 2.30)	0.29	(0.04, 2.30)	0.26	(0.03, 2.10)	0.26	(0.03, 2.12)
Divorced	1.47	(0.79, 2.75)	1.48	(0.79, 2.76)	1.55	(0.83, 2.92)	1.55	(0.83, 2.92)
Widowed	1.08	(0.46, 2.52)	1.08	(0.46, 2.54)	1.06	(0.46, 2.48)	1.06	(0.45, 2.47)
Single/never married	0.75	(0.47, 1.18)	0.75	(0.47, 1.19)	0.76	(0.48, 1.21)	0.76	(0.48, 1.21)
Do not know/refused	0.00	(0.00, 1000.00)	0.00	(0.00, 1000.00)	0.00	(0.00, 1000.00)	0.00	(0.00, 1000.00)
Insurance								
Private	1		1		1		1	
Medicare	1.19	(0.67, 2.10)	1.18	(0.67, 2.10)	1.20	(0.67, 2.13)	1.20	(0.68, 2.13)
Medicaid	1.03	(0.58, 1.84)	1.02	(0.57, 1.83)	1.00	(0.56, 1.80)	1.01	(0.56, 1.81)
Self-pay	0.57	(0.16, 2.10)	0.56	(0.15, 2.06)	0.57	(0.15, 2.08)	0.56	(0.15, 2.06)
Other	2.12	(0.98, 4.57)	2.10	(0.97, 4.54)	**2.21**	**(1.02, 4.78)**	**2.25**	**(1.04, 4.87)**
None	0.43	(0.18, 1.02)	0.43	(0.18, 1.02)	0.46	(0.19, 1.11)	0.46	(0.19, 1.12)
Do not know/refused	0.71	(0.24, 2.10)	0.71	(0.24, 2.10)	0.68	(0.23, 2.04)	0.69	(0.23, 2.05)
Season								
Winter (vs. summer)	0.77	(0.54, 1.11)	0.77	(0.54, 1.11)	0.78	(0.54, 1.13)	0.78	(0.54, 1.13)
Sampling Frame								
Cell (reference)	1		1		1		1	
Landline	1.05	(0.51, 2.16)	1.05	(0.51, 2.16)	1.07	(0.52, 2.21)	1.08	(0.52, 2.22)
Online	0.58	(0.29, 1.16)	0.58	(0.29, 1.16)	0.60	(0.30, 1.20)	0.59	(0.29, 1.19)

**Bolded** values indicates statistical significance.

**Table 4 ijerph-20-06648-t004:** Exposures and Self-Reported General Health.

	Model 2.1: NO_2_		Model 2.2: PM_2.5_		Model 2.3: Perceived Pollution		Model 2.4: All Exposures	
	OR	(95% CI)	OR	(95% CI)	OR	(95% CI)	OR	(95% CI)
Potential Exposures of Interest								
NO_2_	0.97	(0.70, 1.34)	-	-	-	-	0.71	(0.31, 1.61)
PM_2.5_	-	-	1.08	(0.33, 3.59)	-	-	3.42	(0.17, 68.82)
Perceived pollution (PP)	-	-	-	-	1.01	(0.93, 1.09)	1.01	(0.93, 1.09)
Self-Reported Health								
Asthma “Yes”	**0.52**	**(0.34, 0.81)**	**0.52**	**(0.34, 0.81)**	**0.52**	**(0.34, 0.81)**	**0.52**	**(0.34, 0.81)**
Mental health condition “Yes”	**0.47**	**(0.29, 0.78)**	**0.47**	**(0.29, 0.78)**	**0.48**	**(0.29, 0.78)**	**0.48**	**(0.29, 0.79)**
Depression “Yes”	**0.54**	**(0.34, 0.84)**	**0.53**	**(0.34, 0.84)**	**0.53**	**(0.34, 0.84)**	**0.53**	**(0.34, 0.84)**
Gender								
Male (reference)	1		1		1		1	
Female	0.93	(0.66, 1.33)	0.93	(0.66, 1.33)	0.93	(0.66, 1.33)	0.92	(0.65, 1.32)
Do not know/refused	**0.18**	**(0.03, 0.99)**	0.18	(0.03, 1.02)	0.18	(0.03, 1.02)	0.18	(0.03, 1.03)
Race								
White (reference)	1		1		1		1	
Black	1.06	(0.67, 1.67)	1.06	(0.67, 1.67)	1.06	(0.67, 1.67)	1.05	(0.67, 1.66)
Asian	**0.50**	**(0.27, 0.92)**	**0.50**	**(0.28, 0.92)**	**0.50**	**(0.27, 0.92)**	**0.52**	**(0.28, 0.95)**
Native	1.96	(0.41, 9.35)	1.96	(0.41, 9.31)	1.95	(0.41, 9.31)	1.90	(0.40, 9.05)
Other	0.65	(0.34, 1.23)	0.65	(0.34, 1.23)	0.65	(0.34, 1.23)	0.64	(0.34, 1.21)
Do not know/refused	1.09	(0.27, 4.38)	1.09	(0.27, 4.40)	1.09	(0.27, 4.40)	1.04	(0.26, 4.20)
Hispanic/Latino								
No (reference)	1		1		1		1	
Yes	1.16	(0.69, 1.96)	1.16	(0.69, 1.95)	1.16	(0.69, 1.95)	1.14	(0.67, 1.92)
Do not know/refused	1000.00	(0.00, 1000.00)	1000.00	(0.00, 1000.00)	1000.00	(0.00, 1000.00)	1000.00	(0.00, 1000.00)
Numeric Variables								
Age (years)	**0.98**	**(0.97, 1.00)**	**0.98**	**(0.97, 1.00)**	**0.98**	**(0.97, 1.00)**	**0.98**	**(0.97, 1.00)**
Length of time in neighborhood (decades)	0.92	(0.75, 1.12)	**0.91**	(0.75, 1.11)	0.91	(0.75, 1.11)	0.92	(0.76, 1.13)
Household size	1.07	(0.93, 1.23)	1.07	(0.93, 1.23)	1.07	(0.93, 1.23)	1.07	(0.93, 1.23)
Education								
Less than 8th (reference)	1		1		1		1	
Some high school	3.35	(0.69, 16.36)	3.42	(0.70, 16.68)	3.38	(0.69, 16.47)	3.36	(0.69, 16.32)
High school/GED	**4.77**	**(1.11, 20.52)**	**4.88**	**(1.13, 20.99)**	**4.83**	**(1.13, 20.68)**	**4.80**	**(1.12, 20.55)**
Trade/vocational/nursing	2.79	(0.50, 15.73)	2.87	(0.51, 16.16)	2.84	(0.51, 15.92)	2.86	(0.51, 16.03)
Some college	3.09	(0.73, 13.17)	3.14	(0.74, 13.39)	3.12	(0.73, 13.28)	3.12	(0.74, 13.22)
Associate’s	3.20	(0.70, 14.66)	3.27	(0.71, 14.96)	3.23	(0.71, 14.75)	3.21	(0.70, 14.62)
Bachelor’s	**6.29**	**(1.43, 27.60)**	**6.33**	**(1.44, 27.79)**	**6.31**	**(1.44, 27.70)**	**6.36**	**(1.46, 27.75)**
Master’s	**11.03**	**(2.35, 51.79)**	**11.15**	**(2.38, 52.34)**	**11.08**	**(2.36, 52.02)**	**11.11**	**(2.38, 51.82)**
Professional’s	**9.46**	**(1.42, 63.01)**	**9.54**	**(1.43, 63.50)**	**9.55**	**(1.43, 63.63)**	**9.27**	**(1.40, 61.50)**
Doctoral	**10.05**	**(1.18, 85.61)**	**10.20**	**(1.19, 87.14)**	**10.14**	**(1.19, 86.49)**	**10.32**	**(1.21, 87.86)**
Do not know/refused	5.24	(0.26, 103.95)	5.34	(0.27, 105.71)	5.26	(0.27, 103.81)	5.34	(0.27, 106.54)
Income								
Less than USD 23,000	1		1		1		1	
USD 23,000–USD 45,999	1.18	(0.74, 1.89)	1.19	(0.74, 1.89)	1.19	(0.74, 1.90)	1.20	(0.75, 1.91)
USD 46,000–USD 69,999	1.39	(0.78, 2.48)	1.40	(0.78, 2.50)	1.40	(0.78, 2.51)	1.42	(0.79, 2.54)
USD 70,000–USD 92,999	1.20	(0.64, 2.23)	1.20	(0.65, 2.23)	1.21	(0.65, 2.24)	1.19	(0.64, 2.22)
USD 93,000–USD 135,000	1.49	(0.71, 3.12)	1.50	(0.71, 3.15)	1.51	(0.71, 3.18)	1.50	(0.71, 3.17)
USD 135,000+	**3.28**	**(1.12, 9.59)**	**3.26**	**(1.12, 9.53)**	**3.29**	**(1.12, 9.65)**	**3.29**	**(1.12, 9.66)**
Do not know/refused	1.09	(0.38, 3.14)	1.09	(0.38, 3.12)	1.10	(0.38, 3.15)	1.10	(0.38, 3.16)
Marital Status								
Married/living as married	1		1		1		1	
Separated	1.39	(0.37, 5.26)	1.38	(0.36, 5.26)	1.38	(0.36, 5.24)	1.38	(0.36, 5.24)
Divorced	1.55	(0.82, 2.95)	1.55	(0.81, 2.94)	1.55	(0.82, 2.96)	1.55	(0.81, 2.94)
Widowed	1.53	(0.71, 3.30)	1.53	(0.71, 3.31)	1.53	(0.71, 3.30)	1.53	(0.71, 3.31)
Single/never married	1.02	(0.66, 1.58)	1.02	(0.66, 1.58)	1.02	(0.66, 1.59)	1.02	(0.66, 1.59)
Do not know/refused	0.73	(0.06, 8.48)	0.72	(0.06, 8.30)	0.71	(0.06, 8.26)	0.77	(0.06, 9.20)
Insurance								
Private	1		1		1		1	
Medicare	**0.59**	**(0.35, 0.97)**	**0.59**	**(0.35, 0.98)**	**0.59**	**(0.35, 0.98)**	**0.58**	**(0.35, 0.97)**
Medicaid	0.68	(0.39, 1.17)	0.68	(0.39, 1.18)	0.68	(0.39, 1.17)	0.66	(0.38, 1.15)
Self-pay	0.77	(0.24, 2.47)	0.76	(0.24, 2.46)	0.76	(0.24, 2.47)	0.73	(0.23, 2.37)
Other	0.78	(0.33, 1.87)	0.78	(0.33, 1.88)	0.79	(0.33, 1.89)	0.78	(0.32, 1.87)
None	**0.36**	**(0.20, 0.65)**	**0.36**	**(0.20, 0.65)**	**0.36**	**(0.20, 0.65)**	**0.37**	**(0.21, 0.65)**
Do not know/refused	1.11	(0.35, 3.51)	1.11	(0.35, 3.53)	1.11	(0.35, 3.52)	1.11	(0.35, 3.53)
Season								
Winter (vs. summer)	**1.47**	**(1.04, 2.07)**	**1.47**	**(1.04, 2.07)**	**1.47**	**(1.04, 2.07)**	**1.48**	**(1.05, 2.09)**
Sampling frame								
Cell	1		1		1		1	
Landline	0.48	(0.20, 1.14)	0.48	(0.20, 1.14)	0.49	(0.21, 1.15)	0.48	(0.20, 1.13)
Online	0.50	(0.22, 1.16)	0.50	(0.22, 1.16)	0.50	(0.22, 1.16)	0.50	(0.22, 1.16)

**Bolded** values indicates statistical significance.

## Data Availability

Survey data can be made available upon request to the corresponding author. NYCCAS data may be available through NYC Department of Health and Mental Hygiene.

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
