# Peer review of "Perception Matters: Perceived vs. Objective Air Quality Measures and Asthma Diagnosis among Urban Adults"

_ijerph, 2023, doi:10.3390/ijerph20176648_

Round 1

Reviewer 1 Report

The authors may consider to include references of studies from other parts of the world in the introduction section. It will be easier for readers to understand if the names of other places/countries would be mentioned in the sentences.

There is no robust statistical method used in this study. The data is presented in detail however the analysis part is quite small to justify the publication. Authors may consider to apply robust methods to analyze the data sets presented in the manuscript.

The discussion section comprise of the general or overall discussion of results, however it require to discuss the specific results of the study as well in the light of the previous studies and for their own significance too.

Author Response

Reviewer #1

The authors may consider to include references of studies from other parts of the world in the introduction section. It will be easier for readers to understand if the names of other places/countries would be mentioned in the sentences.

  • Thank you for this suggestion. It has been clarified where some of the key studies were performed, and some international references are included.

There is no robust statistical method used in this study. The data is presented in detail however the analysis part is quite small to justify the publication. Authors may consider to apply robust methods to analyze the data sets presented in the manuscript.

  • It is not clear what the reviewer is referring to as a “robust method”? We have used logistic regression with extensive adjustment for potential confounders and exploration of key mediators, along with key sensitivity analyses, including using alternative model forms (Proc GLM), varying resolution of confounder metrics (e.g., categorical vs. continuous, where possible), vary the threshold for mental health outcomes and composition of mental health challenge variable, and using categorical forms of outcome metrics where possible.

  • The analysis is indeed a very simple one – aimed to present the clear distinction we found, in our data, and in the apparent strength of association for these two ways of measuring and understanding air pollution effects on health, pointing towards the need for further studies to be better designed to disentangle these two potential pathways.

The discussion section comprise of the general or overall discussion of results, however it require to discuss the specific results of the study as well in the light of the previous studies and for their own significance too.

  • Thank you for this suggestion. Further detail has been added in the Discussion section.

Reviewer 2 Report

Overall, I think the topic of this paper is very interesting: how does perception of air quality relate to objective air quality? However, the overall design can be improved.

1. Some important factors need to be collected, such as distance to major roads, type of building, etc.

2. I am not sure if participants’ answers to the three questions would accurately reflect their perception of air quality. More information, such as odor, could be collected.

3. It would be difficult to link these three questions with objective air quality, as it would require consideration of different pollutants and a combination thereof.

4. For the association between pollution and health, some key information is missing, such as family history.

Author Response

Reviewer #2

Overall, I think the topic of this paper is very interesting: how does perception of air quality relate to objective air quality? However, the overall design can be improved.

- Thank you for this comment. We have taken these comments into serious consideration, as detailed below.

  1. Some important factors need to be collected, such as distance to major roads, type of building, etc.

- To clarify, these are the source variables that are already accounted for in the NYCCAS air pollution exposure models used.  It would nullify the air pollution-health associations if we were to correct for the sources of air pollution. Those models are extensively detailed in Clougherty et al 2103 and Matte et al, 2013, as cited.   That work was previously performed under the auspices of the NYC Dept of Health and Mental Hygiene, and would not be appropriate to detail as part of this study. We have, however, added some language to (very briefly) describe those models.

  1. I am not sure if participants’ answers to the three questions would accurately reflect their perception of air quality. More information, such as odor, could be collected.

- Thank you for this comment. There are few validated measures of perceived air quality available in the current literature.  We have opted to use one of these few validated scale (cite), but, hopefully, novel results such as ours will support the call for more serious consideration of perceived air quality in the modern environmental health literature, and for more extensive scales to be developed and validated.

  1. It would be difficult to link these three questions with objective air quality, as it would require consideration of different pollutants and a combination thereof.

- Correct. The scale captures overall perceived “air quality,” not specific sources, as it is designed to capture the residents’ lived experience. This is the reason why we chose to compare against two difference ambient air pollutants which are shown to have somewhat different spatial patterning within NYC.

  1. For the association between pollution and health, some key information is missing, such as family history.

- There are an enormous number of factors which may affect health or air pollution susceptibility, including family history and genetics, but, in order to plausibly confound the association, these would need be differentially distributed with respect to both the confounder and the outcome.

Here, because we are comparing models that use objective vs. perceived exposure metrics (vs. both), a potential confounder would need be differently associated with the perceived vs. objective metrics, in order to meaningfully alter their relative associations with health. What factors could modify one but not the other?  Perhaps odor sensitivity?  This is not commonly-collected data, but would be interesting to collect and explore, if future studies are designed from the outset to consider both perceived and objective air quality.

We have indeed adjusted for many of the likely potential confounders of this relationship, as far as our data allows, as listed in Tables 1, and now in the test, including: sex, race/ ethnicity, age, education, insurance status, income, marital status, family size, neighborhood tenure, season, and survey frame.  Models were sensitivity-tested by, among other approaches, varying the resolution of confounder metrics (e.g., categorical vs. continuous, where possible).

Reviewer 3 Report

The article analyzed the relationships between perceived and objective pollution exposure metrics and health outcomes. The findings revealed that perceived air quality exhibited a stronger association with asthma status compared to objective air quality. However, no significant association was observed between perceived air quality and overall self-reported general health. The study underscores the significance of considering both physical and psychosocial aspects of air pollution to comprehend its health impacts. While this study is intriguing and demonstrates novelty, there is room for refinement in terms of both the structural organization and content of the article.

The results and discussion sections should be the focal points for readers. However, the current article merely provides a concise description in these sections. The author simply lists the results without offering further elaboration or explanation.

(1)While discussing the association between self-reported asthma and overall self-rated health, it is worth exploring the underlying reasons for this disparity and investigating why there is a stronger association between self-reported asthma and perceived air quality.

(2)In the discussion of research results, a more detailed comparison and discussion of previous research findings should be provided to emphasize the contribution of this study to existing research.

(3)When conducting data analysis on objective and perceived air pollution exposure, it is important to specify the location of the data mentioned in the description. This makes it easier for readers to relate the content to the tabular data (e.g., Line 247 to 252).

(4)Enhancing methodological details: there is no description of the statistical analysis methods and software tools used. In the data analysis section, only the use of Pearson correlation coefficient and logistic regression analysis is mentioned, but there is no explanation of how adjustments were made, variable selection was performed, or how the results were interpreted.

(5)Other details:

Line 143-152: This part of the content seems inappropriate for inclusion in the article and should be deleted.

Line 131: The highly-refined land-use use regression models of the New York City Community Air Survey is mentioned, but it is not introduced in the Materials and Methods section.

The English language in the provided manuscript is generally good.

Author Response

The article analyzed the relationships between perceived and objective pollution exposure metrics and health outcomes. The findings revealed that perceived air quality exhibited a stronger association with asthma status compared to objective air quality. However, no significant association was observed between perceived air quality and overall self-reported general health. The study underscores the significance of considering both physical and psychosocial aspects of air pollution to comprehend its health impacts. While this study is intriguing and demonstrates novelty, there is room for refinement in terms of both the structural organization and content of the article.

- Thank you for this comment. We have taken these comments into serious consideration, as detailed below.

The results and discussion sections should be the focal points for readers. However, the current article merely provides a concise description in these sections. The author simply lists the results without offering further elaboration or explanation.

  • Thank you for this suggestion. We wanted to be a bit careful to not over-interpret results in the Results section, but have added some additional detail there, and have added further detail in the Discussion section.

(1) While discussing the association between self-reported asthma and overall self-rated health, it is worth exploring the underlying reasons for this disparity and investigating why there is a stronger association between self-reported asthma and perceived air quality.

- Thank you for this suggestion. Our hypothesis, now further emphasized in the text, is that asthma/ respiratory distress (in contrast to overall self-rated health) is more commonly understood to be impacted by air quality. As such, poor perceived air quality may be more likely to trigger (perceived or “real” respiratory distress in individuals, in contrast to other forms of ill health.

(2) In the discussion of research results, a more detailed comparison and discussion of previous research findings should be provided to emphasize the contribution of this study to existing research.

  • Thank you for this suggestion. Further detail to this point has been added in the Discussion section.

(3) When conducting data analysis on objective and perceived air pollution exposure, it is important to specify the location of the data mentioned in the description. This makes it easier for readers to relate the content to the tabular data (e.g., Line 247 to 252).

  • - Thank you for this suggestion. It has been clarified where some of the key studies were performed, as described above.

(4) Enhancing methodological details: there is no description of the statistical analysis methods and software tools used. In the data analysis section, only the use of Pearson correlation coefficient and logistic regression analysis is mentioned, but there is no explanation of how adjustments were made, variable selection was performed, or how the results were interpreted.

- Thank you. The Methods section has now been more extensively detailed.

(5) Other details:

Line 143-152: This part of the content seems inappropriate for inclusion in the article and should be deleted.

  • Thank you for catching this copy-paste error we made in using the IJERPH template.

Line 131: The highly-refined land-use use regression models of the New York City Community Air Survey is mentioned, but it is not introduced in the Materials and Methods section.

  • Thank you. As described above, the NYCCAS work was previously performed under the auspices of the NYC Dept of Health and Mental Hygiene, and would not be appropriate to detail as part of this study. We have, however, added some language to (very briefly) describe those models.

Round 2

Reviewer 1 Report

The authors have addressed all the review comments. Therefore, the manuscript is suitable for publication.